# Neither Valid nor Reliable?
# Investigating the Use of LLMs as Judges

**Khaoula Chehbouni**[1,2]    **Mohammed Haddou**[3]    **Jackie Chi Kit Cheung**[1,2]    **Golnoosh Farnadi**[1,2]

[1]McGill University
[2]Mila - Quebec AI Institute
[3]Statistics Canada

## Abstract

Evaluating natural language generation (NLG) systems remains a core challenge of natural language processing (NLP), further complicated by the rise of large language models (LLMs) that aim to be general-purpose. Recently, large language models as judges (LLJs) have emerged as a promising alternative to traditional metrics, but their validity remains underexplored. This position paper argues that the current enthusiasm around LLJs may be premature, as their adoption has outpaced rigorous scrutiny of their reliability and validity as evaluators. Drawing on measurement theory from the social sciences, we identify and critically assess four core assumptions underlying the use of LLJs: their ability to act as proxies for human judgment, their capabilities as evaluators, their scalability, and their cost-effectiveness. We examine how each of these assumptions may be challenged by the inherent limitations of LLMs, LLJs, or current practices in NLG evaluation. To ground our analysis, we explore three applications of LLJs: text summarization, data annotation, and safety alignment. Finally, we highlight the need for more responsible evaluation practices in LLJs evaluation, to ensure that their growing role in the field supports, rather than undermines, progress in NLG.

## 1 Introduction

Evaluating natural language generation (NLG) systems remains a significant challenge—particularly with the advent of "general-purpose" models—due to several factors, including the subjectivity of the task and the high cost of evaluation. The stakes of this challenge are high: metrics and benchmarks not only shape our understanding of model capabilities, but also influence which research space receives attention and funding, ultimately steering the broader trajectory of the field [75]. In this context, researchers have explored the use of large language models as judges (LLJs) as an human-like and cost-effective evaluation metrics [55, 52]. This promising potential has sparked a surge of interest in the use of LLMs as evaluators, and thousands of related papers have appeared on academic research platforms, reflecting the rapid growth and attention the topic is receiving across the research community.

Despite their growing adoption, the *validity* of LLJs remains relatively underexplored [111, 36], with existing work focusing instead on their *reliability* by exploring their consistency over multiple evaluations or robustness to small stylistic changes in prompts [117, 61, 81, 104, 51]. *Validity* and *reliability* are key concepts from *measurement theory*—a social science framework used to inform evaluation practices [1]. In the ML context, researchers have advocated for applying this framework to improve and systematize existing evaluation practices [47, 102, 115]. Similarly, in this position paper, we leverage a measurement theory framework [47] to investigate the key underlying assumptions behind the widespread use of LLJs: (1) their ability to serve as proxies for human evaluators; (2) their capabilities as evaluators; (3) their potential for scalability; and (4) their cost-effectiveness.

39th Conference on Neural Information Processing Systems (NeurIPS 2025) Position Paper Track.

For each of these assumptions, we highlight current limitations—whether inherent to LLMs, related to their use as evaluators, or stemming from existing practices in the NLG community—that may compromise their reliability and validity. While we offer a high-level perspective on the field, we also examine three popular LLJs applications, to ground our analysis in concrete examples: text summarization, data annotation, and safety alignment. We conclude by emphasizing the need for the community to establish robust standards for the responsible evaluation of NLG systems, as well as the importance of accounting for contextual factors during evaluation. These steps are crucial to unlocking the full potential of LLJs, which mark a significant shift in evaluation practices and open promising pathways toward broader, more comprehensive, and more realistic evaluation of LLMs.

**To summarize, this position paper advocates for more rigorous evaluation practices for LLJs, highlighting that their rapid and widespread adoption may have occurred prematurely, without proper evaluation of their reliability and validity as evaluators.**

## 2   Large Language Models as Judges

Early work on LLJs demonstrated their potential for NLG evaluation [103, 63, 50], sparking growing interest in the research community. Building on these early insights, the adoption of LLJs has quickly proliferated, establishing them as a common tool for evaluation and guidance in a variety of machine learning settings. Leveraging zero-shot or few-shot prompting, LLJs can evaluate outputs across various criteria—such as relevance, fluency, or safety— and can even generate explanations to justify their assessments or provide feedback.

Li et al. [55] formalize the evaluation process of LLJs as an evaluation function that takes various inputs. The evaluation function can be a single LLJ, multiple LLJs (usually referred to as Juries), or a LLJ with a human in the loop. This function takes as input: an *evaluation type*: pointwise (one item at the time), pairwise or listwise; an *evaluation criteria*: specific to the task at hand and refers to linguistic quality, content accuracy or task-specific metrics; an *evaluation item*; an *optional reference*: the evaluation can be reference-based or reference-free. This evaluation function can produce three outputs: the *evaluation result*: e.g, a score, ranking, label or qualitative assessment; the *explanation*: an explanation of the reasoning that led to the assessment; the *feedback*: suggestions to improve the input. This formalization accounts for the high variety of LLJ paradigms, we refer to Li et al. [55]'s work on the topic for a comprehensive survey on LLJs. Furthermore, while different types of LLJs may exhibit different biases, our analysis in this paper deliberately adopts a higher-level perspective. This enables us to provide a broader perspective on their use and introduction in the field, rather than limiting the discussion to biases tied to specific evaluation types.

Although early work on LLJs introduced them as an alternative to traditional NLG evaluation metrics [103, 63, 50], their use has since expanded to different applications. Li et al. [55] identify three main functionalities: (1) performance evaluation, (2) model enhancement, and (3) data construction. Performance evaluation refers to the conventional use of LLJs to assess model outputs or overall performance. Model enhancement captures the use of LLJs to improve models, either through reward modeling or by providing feedback throughout the training process. Lastly, data construction involves leveraging LLJs for tasks such as data annotation or data generation. In this work, we examine the use of LLJs along three use cases to ground our analysis, each corresponding to one of these functionalities: text summarization, safety alignment, and data annotation.

## 3   Measurement Theory

In this section, we briefly introduce measurement theory, which offers a conceptual framework to formalize and evaluate the validity and reliability of an evaluation [111]. Measurement theory has its roots in the quantitative social sciences, particularly in fields such as psychology, education, and political science. Scholars in these disciplines have long aimed to develop formal methods for validating theoretical—and often abstract—concepts. Adcock and Collier [1] outline a four-level framework to understand the connection between concepts and observations. At the most abstract level is the background concept, encompassing the full range of meanings a concept might have. This is followed by the systematized concept, which refers to the precise definition adopted by researchers for analytical purposes. The third level involves the measures, or the scoring procedures used to assess the concept. Finally, the fourth level consists of the scores, which are derived from applying

Table 1: Components of construct validity as described by Jacobs and Wallach [47]

| Dimension | Description |
| --- | --- |
| *Face Validity* | The measurements obtained from the evaluation look plausible. |
| *Content Validity* | The operationalization captures all relevant aspects of the underlying construct. |
| *Convergent Validity* | The measurements obtained correlate with other measurements of the construct. |
| *Discriminant Validity* | The measurements variation suggests the operationalization captures other constructs. |
| *Predictive Validity* | The measurements are predictive of relevant observable properties related to the construct. |
| *Hypothesis Validity* | The measurements support known hypotheses about the construct. |
| *Consequential Validity* | The consequences of using the measurements obtained from an evaluation. |

the measures. In this context, *conceptualization* refers to the process of refining a background concept into a systematized concept, while *operationalization* involves converting the systematized concept into specific indicators or procedures for generating scores.

According to Adcock and Collier [1], a measurement is considered *valid* when the resulting observations—or scores—accurately reflect the content of the systematized concept. Validity is often discussed in light of measurement error, which refers to the difference between the observed score and the systematized concept it aims to represent. Measurement errors can be either systematic or random. Systematic errors, associated to bias, pertain to issues of *validity*, whereas random errors, manifesting as inconsistent results across repeated applications of the same procedure, relate to *reliability* [1].

Building on these foundations, Jacobs and Wallach [47] introduce a framework for understanding fairness in computational systems with respect to *construct reliability* and *construct validity* [18]. Construct reliability is defined in terms of *test-retest reliability*, i.e. how stable the scores obtained from the measurement model are over time, while construct validity is understood as being both context-dependent [1] and gradational in nature [47] and decomposed into seven dimensions described in Table 1.

## 4    Investigating the Assumptions Behind the Use of LLMs As Judges

Whether in the conceptualization or operationalization of a construct, the process of measurement modeling inevitably involves underlying assumptions [47]. To inform our position, we conducted a high-level qualitative review of commonly cited works on LLJs, focusing on how researchers motivate their use and describe their applications. The goal was to surface frequently recurring themes, implicit assumptions, and shared framings that appear across papers. Although not exhaustive, this approach provides insight into the dominant narratives and foundational premises that shape current research directions. We first explored the literature on LLJs across various applications (text summarization, machine translation, alignment, etc.) to get a broader understanding of their use, before focusing more in-depth on three applications illustrating the different functionalities presented in Section 2. As we believe that assessing the validity and reliability of LLJs needs to be done in context.

In this section, we identify and examine four common assumptions motivating the use of LLJs in the field. For each assumption, we illustrate key challenges using concrete examples from existing literature. Table 2 presents example quotes for each of these assumptions while Figure 1 presents an overview of our findings.

### 4.1    Assumption 1: LLMs as a Proxy for Human Judgment

Traditionally, NLG evaluation has relied on human annotators to assess the quality of language model outputs. Consequently, a range of benchmarks has been developed for tasks such as summarization, data-to-text generation, and machine translation, incorporating "human judgment"—either in the form of a score or human-written reference outputs. However, recent advances in LLMs are beginning to shift this paradigm. In particular, progress in reinforcement learning from human feedback [76] has significantly enhanced LLMs' ability to produce human-like text, making it increasingly challenging to distinguish between human-written and LLM-generated content [17]. The ability of LLMs to generate human-like text and align with human preferences [76] has prompted researchers to investigate their potential as effective alternatives to humans in NLG evaluation.

Table 2: Example quotes illustrating the assumptions behind the use of LLMs judges from selected publications.

**LLMs as a Proxy for Human Judgment**

*Zheng et al. [117]: To automate the evaluation, we explore the use of state-of-the-art LLMs, such as GPT-4, as a surrogate for humans.*
*Gilardi et al. [33]: It strongly suggests that ChatGPT may already be a superior approach compared to crowd annotations on platforms such as MTurk.*

**LLMs as Capable Evaluators**

*Chiang and Lee [14]: LLMs have demonstrated exceptional performance on unseen tasks when only the task instructions are provided.*
*Mohta et al. [70]: It also exhibits the unique capability to provide reasons for classification, a feature often absent in human-labeled data.*

**LLMs as Scalable Evaluators**

*Sun et al. [95]: However, acquiring high-quality human annotations, including consistent response demonstrations and in-distribution preferences, is costly and not scalable.*
*Mazeika et al. [68]: However, companies currently rely on manual red teaming, which suffers from poor scalability.*

**LLMs as Cost-Effective Evaluators**

*He et al. [39]: From a cost perspective, GPT-4 is also more affordable than hiring MTurk workers.*
*Sun et al. [96]: [...] offering cost-effective and accessible alternatives.*

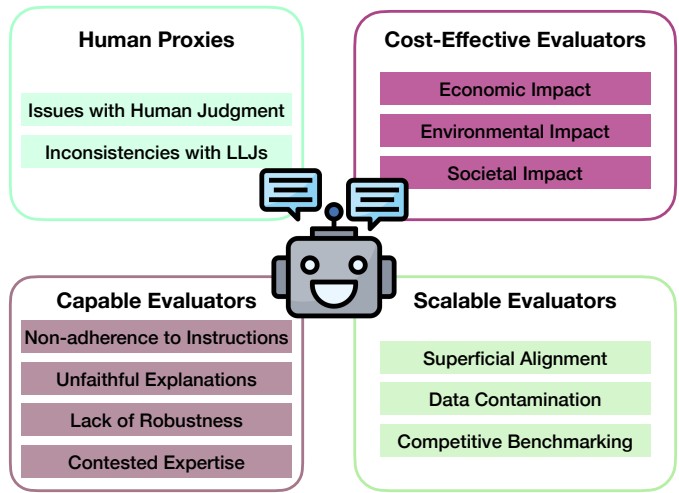

Figure 1: We look at the main assumptions made in the LLJs literature and identify potential pitfalls that can undermine the validity and reliability of LLMs as measurement models.

Early studies on LLJs [50, 14, 103, 63] have primarily validated their use through *convergent validity*. As shown in Table 1, this approach assumes that a metric is valid if it correlates with an existing, already validated metric for the same construct [47]. As a result, researchers have evaluated LLM-generated text using other LLJs, comparing their scores to human gold standards present in common NLG benchmarks (e.g., `SummEval` [27], `RealSumm` [8] or `Topical-Chat` [69]) using correlation metrics such as Pearson, Spearman, or Kendall's Tau. Demonstrating correlation between human ground-truth judgments and LLM-based evaluations has laid the groundwork for the field, driving their widespread adoption across various tasks and applications in academia and industry alike [52]. Although a body of work [63, 50] has shown a degree of correlation between LLM-based evaluations

and human judgment—indicating a form of *convergent validity*—we argue that existing shortcomings in NLG evaluation practices undermine the validity of LLJs as measurement models because the human judgments themselves might not be valid. In this section, we highlight how these shortcomings may undermine LLJs *convergent validity*.

**Inconsistencies in Human Judgment Collection.**   As noted by Nenkova and Passonneau [74] : *"to show that an automatic method is a reasonable approximation of human judgments, one needs to demonstrate that these [human judgments] can be reliably elicited."* However, prior work [120, 42] has revealed significant inconsistencies in how human judgments in NLG evaluation are being elicited and collected, casting doubt on their validity as a benchmark. Indeed, while human evaluation has long been considered the gold standard in NLG, there remains little agreement on what constitutes human judgment or how it should be collected. Howcroft et al. [42] examine twenty years of human evaluation practices in NLG and reveal a lack of shared practices within the community. They highlight significant inconsistencies in the definitions of evaluation criteria, when such definitions are provided at all, along with vague instructions for annotators (e.g., missing examples or clear rating scales) which lead to diverse interpretations of the evaluation task. Resulting in a clear mismatch between the evaluation criteria intended by researchers and how annotators actually understand them [17]. In addition, Zhou et al. [120] interview both academic and non-academic NLG practitioners about their evaluation practices, and reveal additional pitfalls in human evaluation practices, including ambiguity around the kind of expertise needed when involving human annotators, a conflation of quality criteria with their measurement, and the re-purposing of benchmarks created for other tasks. These issues have increasingly called into question the role of human judgment as the gold standard in NLG evaluation, sparking concerns about quality and reproducibility. This raises doubts about whether human judgments capture the intended aspects of evaluation.

**Inconsistencies in LLJs Judgment Collection.**   Despite this uncertain foundation, the LLJs literature has adopted correlation with human judgment as the primary validation criterion of their use without critically investigating what aspects of the human judgment construct LLJs actually correlate with. Even this correlation is sometimes disputed—either because practitioners question whether LLJs actually correlate with human judgment as they notice great variability across tasks and benchmarks [51, 6], or because the methods used to compute the correlation are themselves contested [25]. For example, Elangovan et al. [25] show how human uncertainty in labeling affects the correlation scores produced by an LLJ. Specifically, under high human uncertainty—such as an improperly documented subjective annotation task—correlation between automatic metrics like LLJs may appear artificially inflated. Moreover, the literature on LLM judges seems to reproduce and even exacerbate many of the same issues found in previous NLG evaluation research—especially inconsistent conceptualization of evaluation criteria and ambiguous operationalization [43], not just across different benchmarks but also within individual benchmarks themselves. These inconsistencies in both human and LLM judgment collection practices raise important concerns about the validity of using LLMs as reliable proxies for human evaluation. Without standardized definitions, evaluation methods, and scoring scales, it becomes difficult to ensure that LLM-based assessments faithfully reflect human judgment.

*Example: The SummEval Benchmark.* The `SummEval` benchmark [27] has emerged as a key reference for validating the use of LLJs within the broader community given its pivotal role in NLG evaluation research. To better understand how LLJs judgements are collected and evaluated, we look at how different papers use this benchmark and notice various inconsistencies illustrated in Table 3. For example, while the original work provides the instructions given to annotators for the different evaluation criteria (relevance, consistency, fluency, coherence), work on LLJs do not always make use of these definitions. Across three different papers, only one uses the provided definition of fluency but also includes irrelevant information—confusing disfluency (which refers to a vocal communication disorder) with a lack of fluency in text. One paper does not provide any definition of fluency at all, while another introduces its own interpretation of the criterion. Additionally, although the human evaluation in the benchmark was conducted via relative comparisons by assessing five summaries simultaneously, studies using LLJs adopt either absolute or pairwise comparisons. They also employ varying rating scales (e.g., 1–3 or 1–100) instead of the benchmark's original 1–5 Likert scale.

Table 3: Instructions provided to LLM judges for evaluating the fluency criterion in SummEval [27]

| Instructions | Scale | Process |
|---|---|---|
| **Original instructions to annotators from Fabbri et al. [27]:** This rating measures the quality of individual sentences, are they well-written and grammatically correct. Consider the quality of individual sentences. | Likert Scale (1-5) | Relative Comparison |
| The quality of the summary in terms of grammar, spelling, punctuation, word choice, and sentence structure. 1: Poor. The summary has many errors that make it hard to understand or sound unnatural. 2: Fair. The summary has some errors that affect the clarity or smoothness of the text, but the main points are still comprehensible. 3: Good. The summary has few or no errors and is easy to read and follow [63]. | Likert Scale (1-3) | Absolute Comparison |
| Score the following news summarization given the corresponding news with respect to fluency with one to five stars, where one star means "disfluency" and five stars means "perfect fluency". Note that fluency measures the quality of individual sentences, are they well-written and grammatically correct. Consider the quality of individual sentences[103]. | Likert Scale from 1-5 and from 1-100 | Absolute Comparison |
| Which Summary is more fluent relative to the passage, Summary A or Summary B? or Provide a score between 1 and 10 that measures the summaries' fluency [65]. | Binary Option or Scale 1-10 | Pairwise Comparison or Absolute Comparison |

## 4.2 Assumption 2: LLMs as Capable Evaluators

Another key assumption underlying the use of LLMs as judges is their high potential as evaluators. General-purposes LLMs have demonstrated impressive capabilities in in-context learning and instruction-following [105, 107, 106]—even surpassing human performance in certain tasks [5], suggesting that they could be used as evaluators without additional training. In this section, we explore how inherent limitations in LLMs' capabilities may affect their validity and reliability as evaluators across four key dimensions: instructions adherence, explainability, robustness, and expertise.

**Instruction Adherence.** While LLMs are widely recognized for their strong instruction-following capabilities [76], recent work has exposed notable limitations in these capabilities when applied to NLG evaluation. For instance, Hu et al. [43] show that LLMs frequently rely on their own interpretations of evaluation criteria, rather than following the instructions provided in the prompts, particularly across popular quality criteria used in the NLG literature. Moreover, LLMs often conflate distinct dimensions of evaluation, such as fluency and relevance, raising concerns about their *discriminant validity* as their operationalization of one criterion may inadvertently reflect aspects of another. Feuer et al. [28] corroborate these findings and further show that LLM judges exhibit strong implicit biases across quality criteria, assigning varying levels of importance to each and, as a result, scoring them inconsistently.

**Explainability.** A body of work has explored the effect of self-explanation on LLMs as evaluators, demonstrating that generating rationales through chain-of-thought reasoning can increase the correlation between model-generated scores and human judgments [15, 71, 48, 38], and improve transparency and interpretability of LLMs as judges. However, none of these studies examined the faithfulness of the generated explanations—either omitting any evaluation of the explanations altogether or focusing instead on aspects such as coherence [71], or the consistency and stability of the rationales provided [38]. As a result, they primarily assess the *face validity* of LLMs as judges, rather than rigorously validating their role as interpretable evaluation metrics. This is, in fact, one of the key challenges in LLM validation: their high *face validity*, as their outputs often appear coherent and plausible, even when they are wrong [90, 2].

**Robustness.** While the literature on LLJs has paid limited attention to their *construct validity* (§ 3), focusing primarily on *convergent validity* (§ 4.1) and *face validity* as seen above, it has, by contrast, extensively examined their *construct reliability*, particularly through assessments of *test-retest reliability*, given the known stochasticity of LLMs. For instance, [117, 56, 116, 54, 41, 61, 81, 104, 51, 112] examine position bias (also referred to as selection bias or order bias), showing that LLJs can favor responses based on their position in the response set [55]. Indeed, studies have demonstrated that LLJs are vulnerable to a broad spectrum of biases in their role as evaluators [52]. One prominent category includes cognitive biases [51]. For instance, saliency or verbosity bias describes the tendency

of LLMs to be swayed by the length of responses [117, 112]. Other biases include compassion-fade bias (favoring recognizable names over anonymized aliases), bandwagon-effect bias (favoring majority opinions), and attentional bias (giving too much attention to irrelevant details). We refer readers to Li et al. [55]'s survey for a more comprehensive discussion of the different documented biases in LLJs. Finally, a growing body of research [81, 118, 89, 57, 24] has demonstrated that LLJs are highly susceptible to superficial adversarial attacks and prompt manipulations. Raina et al. [81] show that it is possible to design universal attacks to inflate LLJs scores. Similarly, Li et al. [57] design a simple attack to inflate the scores obtained by diverse state-of-the-art LLJs. In the context of LLMs safety judges, Eiras et al. [24] show that they can lead a model to misclassify up to 100% of harmful generations as harmless through simple prompt variations. This lack of robustness to a multitude of biases and adversarial attacks undermines the *construct reliability* of LLJs, as small perturbations may lead to completely different evaluation results.

**Expertise.** A key argument in the literature supporting the use of LLJs is their strong performance on specific tasks. As Kocmi and Federmann [50] point out: *"if the model can translate, it may be able to differentiate good from bad translations"*. Others may argue that comparison is easier than generation—even if both are probably correlated, and is therefore still valid for tasks for which LLJs are known to have inherent weaknesses, including math and reasoning [117], factuality [21] and safety [24]. However, we maintain that an LLM performance on a given task may impact its *content validity* as a judge for that same task.

*Example: LLJs as Annotators.* Because of LLMs ability to produce human-like text often indistinguishable from human-written text [17] and even perceived as higher quality [114, 91, 78], they are often seen as a great alternative for automated annotation, despite some researchers still disagreeing on their proficiency on the task [51, 72]. Interestingly, LLJs have been proposed as substitutes for humans in highly subjective and contested tasks, such as hate speech detection [44] and political affiliation classification [100]. The contested nature of these tasks makes it challenging to establish the content validity of LLJs as annotators. These constructs are highly subjective [40, 35], and relying on LLJs—who tend to yield higher inter-annotator agreement and present their own inherent biases—risks overlooking the valuable diversity found in human disagreement, which is especially important in these contexts. While disagreement is often treated as a marker of poor annotation quality, studies have shown it can instead provide important insights for subjective tasks [29, 49].

## 4.3 Assumption 3: LLMs as Scalable Evaluators

Another key assumption underlying the use of LLJs lies in their potential for scalability. Since scaling is widely recognized as a major driver of LLM performance [106], recent research has increasingly focused on scaling both datasets and model training. Within this context, LLJs have gained momentum, especially for alignment purposes. Considering the prohibitive cost of reinforcement learning from human feedback— which depends heavily on large amount of high quality human data [76], researchers have explored automated alternatives, leveraging LLJs for automatic red-teaming [79], self-improvement [5], self-alignment [96, 95], and human feedback generation [23], among other things. Most notably, LLJs have been widely used for safety at different steps of the mitigation pipeline: for generating harmless preferences, or annotating human preferences, for safety bench-marking, automatic red-teaming or as online guardrails for example [79, 5, 45, 68, 95]. For each of these tasks, researchers have been leveraging one or multiple LLJs through the safety pipeline. In this section, we show how these new safety pipelines can challenge the *discriminant validity* and *predictive validity* of LLJs.

**Scaling Contamination.** Beyond their "traditional" role in evaluation, LLJs are increasingly being used for model enhancement [55] (see Section 2). They can assume various roles throughout the training pipeline, including data generation and annotation, reward modeling, and verification [5, 95, 96]. While these applications have led to notable improvements in utility, the generalization of these performance gains remains to be rigorously validated, especially as such practices blur the boundary between training and testing. Considering that LLJs are mostly validated using publicly available benchmarks, this raises the issue of data contamination: several studies have shown evidence of memorization of popular benchmarks in various state-of-the-art LLMs [34, 83, 19, 22, 4]. Although the extent to which such contamination may inflate the performance of LLJs on popular benchmarks for NLG evaluation has yet to be thoroughly investigated, adjacents issues have been documented in

the literature. A growing body of research has demonstrated *self-enhancement bias* (also referred to as egocentric or narcissist bias) in LLJs, referring to their tendency to favor and inflate evaluation scores for responses generated by models from the same family [117, 64, 77]. Li et al. [53] further demonstrate that this bias can be exacerbated through the phenomenon of *preference leakage*, a specific form of contamination affecting LLJs. Preference leakage arises when LLMs used for data generation and evaluation in the training pipeline are closely related, which is typically the case in alignment settings, as practitioners use off-the-shelf models or previous versions of the same model as a judge. They show how this issue is especially prevalent in LLJs-based benchmarks, like AlpacaEval [23] and Arena-Hard [60], and that its severity increases with the degree of similarity between the models. Similarly, Li et al. [57] show that preference-based LLJs evaluation can be easily manipulated by adapting the responses of a model to align more closely with the judge. In cases where the evaluated model and the judge belong to the same model family, such adaptation may not even be necessary, as their preferences are already aligned.

**Competitive Benchmarking.** While the aforementioned biases raise important concerns about the use of LLJs for model alignment and benchmarking, even in the absence of such biases, incorporating LLJs inside the training pipeline remains questionable. The issue of competitive benchmarking has gained increasing attention in recent years [75], particularly in light of the rapid advancement of LLM capabilities and considering that in NLP, benchmarks are both testing instrument and testing material [87]. The emergence of various leaderboards and other LLJs powered evaluation framework has accentuated these concerns, as automatic evaluations are prone to negative feedback loop and inflated results. For example, Singh et al. [92] expose critical flaws in current evaluation practices that undermine the validity of one of the most widely used leaderboards for LLM evaluation: `Chatbot Arena` [16]. These flaws include unequal data access favoring proprietary providers such as Google and OpenAI, increased risks of overfitting to the benchmark, and the ability for participants to selectively disclose results or privately remove models from the platform. Numerous studies have demonstrated how easily evaluation frameworks can be manipulated [87, 10, 92], and we argue that automatizing the pipeline can only facilitate such practices as seen in [117, 64, 77, 53]. These biases and malpractices undermine the *predictive validity* of LLJs, as their scores are disproportionately affected by confounding factors unrelated to the task. This issue likely arises from overfitting to benchmarks and optimizing for specific metrics instead of the task itself, highlighting a gap between the intended construct (e.g., safety) and its operationalization (e.g., general framework for automated safety mitigation and evaluation).

**Scaling Superficiality.** Zhou et al. [119] introduced the *Superficial Alignment Hypothesis*, which suggests that LLMs acquire most of their knowledge and capabilities during pre-training, while alignment primarily affects the stylistic format of their outputs. This hypothesis is later supported by Lin et al. [62], who show that the most significant distribution shifts between base and aligned LLMs involve stylistic tokens. While such findings should have prompted critical reflection on the effectiveness of current safety mitigation practices—particularly in light of numerous documented failures in the literature [108] and encouraged greater investment in advancing safety research, they have inadvertently steered the field in a less constructive direction. Specifically, the perception that alignment is largely superficial has led to the belief that it can now be achieved at a lower cost [62], and that human feedback may no longer be necessary, paving the way for the adoption of LLJs as a realistic alternative.

*Example: LLMs as Safety Judges.* In an effort to prevent deployed models from producing harmful content, companies have released LLSJs as real-time safeguards. These judges act as guardrails, evaluating user inputs to determine whether they are appropriate for the base model to respond to. Examples of these models include `Llama Guard` [45, 97], `ShielGamma` [113], and `Guardformer` [73], among others [37, 66, 32, 68, 58]. LLSJs are typically responsible for classifying user inputs and model responses acording to predefined safety-taxonomies. While the exact criteria depend on each company's safety policies, they generally cover similar dimensions such as violence, hate speech, sexual content, weapons, illicit substances, suicide, and criminal activity. Eiras et al. [24] show that LLSJs are highly sensitive to distribution shift and adversarial attacks. Notably, they show how small modifications in outputs can lead LLMs safety judges to misclassify up to 100% of harmful generation as harmless. Similarly, Chen and Goldfarb-Tarrant [13] demonstrate that injecting artifacts into safety-related prompts can mislead LLSJs, revealing an over-reliance on surface-level statistical cues rather than a genuine understanding of safety. For instance, adding an apology to an unsafe

prompt may cause the model to wrongly classify it as harmless. This focus on stylistic markers over substantive safety concerns supports the superficial alignment hypothesis and echoes findings from prior work on the limitations of safety safeguards—such as exaggerated safety behaviors [84] and the ways in which such behaviors can reinforce existing societal biases [10]. Such tendencies call into question the *discriminant validity* of LLSJs, as their vulnerability to superficial interventions suggests a limited understanding of safety concepts, relying instead on loosely correlated proxies—such as a model's refusal to respond to a query.

### 4.4 Assumption 4: LLMs as Cost-Effective Evaluators

Another key assumption underlying the use of LLJs is their potential to serve as a more cost-effective alternative to human evaluation. Although LLM-based evaluation can incur higher costs in certain scenarios [28] such as LLJs-based benchmarks like `Arena-Hard` [60], it is generally more economical overall. Instead of relying on crowdworkers (e.g., via Amazon Mechanical Turk) or domain experts, practitioners can now leverage open- or closed-source LLMs for evaluation. While inference costs vary across models, this approach remains more affordable than human labor, particularly as the operational costs of LLMs continue to decrease [28].

The conversation around LLJs as a cheap, realistic, and scalable alternative echoes the introduction of Amazon Mechanical Turk (AMT) into the research sphere, as it was similarly praised for these qualities [9, 93]. Although the platform was initially introduced as a more diverse, scalable, and cost-effective alternative to traditional human evaluation methods, it has since come under scrutiny. For instance, Marshall et al. [67] highlight a decline in data quality on AMT over time, despite the implementation of numerous mitigation strategies aimed at enhancing the reliability of collected data—such as attention checks, reading comprehension tasks, and pre-screening workers based on specific criteria. Beyond this deterioration in quality, researchers have also raised significant ethical concerns about the platform and crowdwork more broadly, particularly emphasizing its exploitative nature, characterized by extremely low wages, lack of transparency, pronounced power asymmetries, and threats to workers' privacy [30, 80, 86].

The AMT example highlights the importance of considering more than just short-term financial costs when adopting new evaluation methods. While localized short-term economies might seem attractive, they can snowball into larger societal impacts once such practices become established in the field. Similarly, the adoption of LLJs often overlooks long-term implications and non-financial costs, factors that are largely ignored in the literature and rarely discussed in critical depth, despite being crucial to establishing the validity of the framework. Using LLMs as benchmarks not only influences perceived progress in the field but also shapes how research is conducted, how LLMs capabilities are understood, and how we interpret the very constructs these models are purported to measure. Moreover, while LLJs offer clear cost savings for researchers, it remains unclear who will ultimately bear the broader costs of their widespread adoption over time. Below, we examine how the non-financial impacts associated with LLJs may affect their *consequential validity* (see Table 1).

**Economic Impact.** Although early efforts to explore LLMs as an alternative to human annotators [33, 3] have been widely criticized in both the media [99, 109] and academic literature [39], researchers continue to pursue the use of LLJs as annotators [121, 100, 70, 121, 44, 39, 38], raising concerns about the future of crowdworkers—an already vulnerable population [80]—especially as many agree that ongoing advances in LLM research are likely to have disruptive effects on the labor market [98, 46]. While automation may appear as a more "ethical" alternative in certain contexts—such as content moderation, where workers are regularly exposed to traumatic material (e.g., child sexual abuse, bestiality, incest [110])—it is crucial to acknowledge that, despite their precarity, these jobs remain the primary source of income for many. Displacing workers does not resolve the underlying issues. Instead, efforts should focus on improving working conditions and developing alternatives that are grounded in the needs and lived realities of these workers [1].

**Environmental Impact.** Although frequently overshadowed by the environmental costs of training[7, 59], the environmental impact of large language models during inference remains considerable—whether in terms of energy consumption [85], carbon emissions [26], or water usage [59]—and continues to grow as model sizes increase [20]. While there is a growing focus on

---

[1]https://data-workers.org/

more efficient computational methods (e.g., model distillation, sparsification) [85], the LLJs literature continues to favor larger models, as they have been shown to produce more accurate evaluations, sometimes even leveraging "juries", i.e., ensembles of smaller LLMs [101], to improve performance. Few have investigated less resource-intensive alternatives, addressing a gap in the current discourse on evaluating LLJs.

**Societal Impact.** LLJs are not exempt from the well-documented societal biases present in large language models [88, 108, 31]—even if such biases have not yet been extensively studied in this context. NLG systems are known to reinforce social stereotypes and discriminatory patterns, and they are prone to producing hate speech, offensive content, and exclusionary language [108]. When used for evaluation, LLJs have the potential to exhibit fairness-related biases, potentially favoring responses associated with certain demographic groups over others. However, few studies have examined the potential of LLJs to reproduce existing discriminatory patterns [94, 112, 12]. Among these, Ye et al. [112] demonstrate that LLJs exhibit diversity bias, meaning their judgment shift in the presence of identity markers. While Chen et al. [12] show evidence of gender bias in LLJs. Given these findings, it is likely that LLJs reproduce societal biases. This highlights the need for further investigation before deploying them, particularly in high-stakes applications.

## 5    The Path Forward

In this section, we offer recommendations to support the responsible and effective integration of LLJs into evaluation practices.

First, despite the wide range of applications for LLJs, there has been little adaptation in how they are deployed or in how evaluations are designed across different tasks and domains—an oversight that can lead to harmful consequences. For instance, while using LLJs to scale red-teaming efforts can enhance the breadth of evaluation, applying the same approach within a safety mitigation pipeline can result in superficial safety behaviors. In such contexts, more targeted and domain-specific mitigation strategies can be more effective than a trial-and-error approach that does not account for the sociotechnical aspect of safety [11]. Evaluating the role of LLJs as evaluators necessitates a comprehensive approach that considers several critical dimensions, including the task at hand, the goal of the evaluation, and the type of LLJ being used, among other things.

Finally, although mitigating the biases of LLJs has become an active area of research [52], the field urgently requires improved evaluation practices. Recent controversies [82, 92] have exposed how tech companies manipulate existing evaluation frameworks, raising serious concerns about data contamination, competitive benchmarking, and overfitting among other things. Despite the importance of evaluation in ML development, there is a lack of rigorous shared practices among practitioners. While technical artifacts like benchmarks and metrics are commonly shared, methodologies and practices are not. [42, 120, 102] highlight that current practices in NLG evaluation lack standardization and systematization, and as demonstrated in this paper, the adoption of LLJs is no exception. LLJs not only reproduce and exacerbate existing issues, but also introduce new challenges for the community. We need to not only focus on mitigating the individual biases associated with LLJs but also consider how to improve NLG evaluation practices as a whole, and better train ML practitioners for such a complex task. It is perhaps time to move beyond a paradigm where we rely on interested companies to provide transparent and comprehensive evaluation of the products they aim to market, and instead work into putting in place proper mechanisms for transparent, valid and reliable evaluation.

## 6    Conclusion

In this work, we have explored how various pitfalls in NLG evaluation practices and the inherent limitations of LLMs can impact LLJs as evaluators. We argue that fully realizing the potential of LLJs depends on our ability to critically and systematically address these challenges. When properly implemented, LLJs offer a valuable opportunity to advance NLG evaluation—whether by enabling more realistic, interactive, and long-term evaluation pipelines that better reflect real-world usage, or by alleviating the burden of problematic annotation tasks involving harmful or traumatic content for example. Therefore, leveraging LLJs effectively will require a careful balance: improving efficiency without disregarding their broader societal impact.

## 7 Acknowledgements

Funding support for project activities has been partially provided by Canada CIFAR AI Chair, NSERC discovery grant and FRQNT grant. We also express our gratitude to Compute Canada and Mila clusters for their support in providing facilities for our evaluations.

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
