# OpenReview forum: "Neither Valid nor Reliable? Investigating the Use of LLMs as Judges"
_NeurIPS.cc/2025/Position_Paper_Track — NeurIPS 2025 Position Paper Track_

### Official Review · Reviewer_pXGB · 2025-08-11

**Significance:** 4
**Presentation:** 3
**Rating:** 8
**Confidence:** 4

**Summary:**

This position paper argues that the widespread adoption of LLJs for NLG evaluation has been premature, outpacing rigorous validation of their reliability and validity. Drawing on measurement theory from social sciences, the authors critically examine four core assumptions: LLMs as proxies for human judgment, their capabilities as evaluators, their scalability, and cost-effectiveness. Through analysis of three applications, namely text summarization, data annotation, and safety alignment, they demonstrate systematic flaws in current LLJ practices and call for more responsible evaluation standards before further deployment.

**Strengths:**

- Principled theoretical framework using measurement theory provides rigorous foundation for critique
- Comprehensive coverage of LLJ applications across the ML pipeline
- Well documented analysis of inconsistencies in current practices with concrete examples
- Strong literature review
- Clear articulation of the stakes, evaluation practices shape research directions and funding

**Weaknesses:**

- Environmental impact discussion feels somewhat tangential to main arguments
- Some sections could be more concise.

**Questions:**

- Given the documented problems with both human evaluation practices and LLJs, what specific standards or methodologies would you recommend for responsible LLJ validation?
- How do you envision the field transitioning from current practices, taking into consideration the cost benefits in terms of time and money?

**Alternative Position:**

Yes, and alternative positions are well-considered and addressed by the argument

**Author Identification:**

No.

**Context:**

4

**Discussion:**

4

**Ethics:**

["NO or VERY MINOR ethics concerns only"]

**Position:**

Yes, the paper argues for or against a position related to machine learning.

**Support:**

4

**Thoroughness:**

3

---

### Official Review · Reviewer_EPEp · 2025-08-15

**Significance:** 4
**Presentation:** 4
**Rating:** 9
**Confidence:** 4

**Summary:**

The paper argues that the LLM as judges for evaluating natural language generation requires more rigorous evaluation practices, suggesting that we haven't evaluated models on reliability and validity aspects. The authors draw from the social science measurement theory which provides a framework for thinking about evaluation, including validity and reliability aspects. The authors bring into question that common narratives that favor use of LLM-as-judges for evaluations, highlighting how some of these narratives are rooted in assumptions that fail or not fully realized in practice. These include, a) LLMs as a proxy for human judgment b) LLMs as capable evaluators c) LLMs as scalable evaluators and d) LLMs as cost-effective evaluators. The authors discuss the limitations in each of these areas.

**Strengths:**

1. The paper is well motivated and grounded in the framework of social science measurement theory, which provides a great way to think about evaluations.
2. The authors have clearly articulated their position and provide an in-depth discussion of different reasons for using LLM-as-judge for NLG evaluations and provided evidence on how these reasons are not fully realized or have some limitations. The authors have also provided excerpts from the prior work describing how people have framed LLM-as-judge evaluations in their own work.

**Weaknesses:**

1. The authors argue that we need to "put work into putting in place proper mechanisms for transparent, valid and reliable evaluation." However, the discussion could benefit from adding more on how do authors envision NLG evaluations to be conducted in the future? What sort of roadmap we would want to follow in order to take steps towards conducting more rigorous evaluations? Are there avenues that should be prioritized first? Some of the suggestions might require institutional changes, what challenges do authors envision there?

**Questions:**

1. Many of the interdisciplinary/applied computational work often spends a lot of time thinking about evaluations, do authors think we have anything to learn from them?

**Alternative Position:**

Yes, and alternative positions are well-considered and addressed by the argument

**Author Identification:**

No.

**Context:**

4

**Discussion:**

4

**Ethics:**

["NO or VERY MINOR ethics concerns only"]

**Position:**

Yes, the paper argues for or against a position related to machine learning.

**Support:**

4

**Thoroughness:**

4

---

### Official Review · Reviewer_VTif · 2025-08-17

**Significance:** 3
**Presentation:** 3
**Rating:** 6
**Confidence:** 3

**Summary:**

This paper analyzes the growing use of Large Language Models as Judges (LLJs) for evaluating NLG systems. The authors argue that adoption has outpaced scientific validation of their reliability and validity. Based on measurement theory, the paper examines four assumptions: (1) a proxy for human judgment, (2) evaluator capability, (3) scalability, and (4) cost-effectiveness. The authors identify key limitations: inconsistent human benchmarks, deviation from instructions, vulnerability to biases such as position and self-enhancement, and data contamination. The authors argue that future progress depends on developing standardized evaluation practices that go beyond simple bias mitigation.

**Strengths:**

- The paper is well-written and timely. Since LLJs are increasingly used not only for evaluation but also in training, a position paper that critically reflects on this trend is both necessary and important.

- It offers a sharp critique, pointing out the limitations of relying solely on correlation with human judgments as a validation criterion. The discussion of vague definitions and inconsistent evaluation scales is also convincing, as these issues can directly cause unreliable scores.

- The paper's discussion on explainability is important. It correctly identifies a critical gap in the literature: while many studies tout the ability of LLJs to provide rationales, the faithfulness of these explanations is rarely scrutinized. The authors rightly call for a more rigorous evaluation of these generated explanations beyond their 'face validity'.

**Weaknesses:**

- The paper does not clearly distinguish between pairwise evaluation and independent scoring. For instance, position bias is mainly an issue in pairwise setups, whereas SummEval illustrates problems tied to inconsistent definitions in independent scoring. Explicitly separating these settings would make the argument stronger.

- The authors suggest that annotator disagreement may be seen as “valuable diversity.” While this can be true in subjective tasks, the claim should be made more carefully, since such disagreement may also stem from ambiguous criteria or poor task/aspect design. In this context, exploring agreement across a diverse panel of LLJs could be a useful complementary direction.

**Questions:**

This paper already provides sufficient discussion; I have no further questions.

**Alternative Position:**

Yes, and alternative positions are well-considered and addressed by the argument

**Author Identification:**

No.

**Context:**

3

**Discussion:**

3

**Ethics:**

["NO or VERY MINOR ethics concerns only"]

**Position:**

Yes, the paper argues for or against a position related to machine learning.

**Support:**

4

**Thoroughness:**

3

---

### Note · Authors · 2025-08-27

**1-11 Submit Again:**

Definitely yes

**1-1 Submission Process:**

3

**1-2 Next Year:**

I found the position paper track to be a valuable initiative, as it made the conference more accessible to types of work that might not otherwise be valued in the main track. However, I believe it would have been preferable to maintain the same evaluation process as the traditional NeurIPS track—not the same criteria, but the rebuttal phase. Given the inherent subjectivity in evaluating this kind of work, the lack of a rebuttal process can amplify biases in reviewers’ assessments. Position papers are, by nature, more “controversial,” and allowing authors to respond would create space for constructive debate, clarification of their stance, and ultimately stronger contributions. Providing authors with the opportunity to engage, clarify, and refine their arguments would have been highly beneficial.

**1-3 Future Development:**

To follow the NeurIPS timeline with an author-responses discussion while maintaining different evaluation criterias.

**1-4 Interest:**

["Panel discussions with other position paper authors", "Structured debates on controversial topics", "Workshops for developing position papers", "Mentorship programs for early-career researchers"]

**1-5 Thoughtful:**

9

**1-6 Supportive:**

10

**1-7 Technical Aspects Versus Position:**

8

**1-8 Gate Keeping:**

10

**1-9 Camera Ready Changes:**

Following the feedback received from the reviewers, we have improved the writing and presentation of our paper to make certain points more explicit (e.g. the necessity of having a discussion around environmental impact and how it relates to the validity of a measurement tool, the switch from low-resource metrics to LLMs, the importance of annotators’ disagreement when evaluating bias but also how it is being used in the field as a proxy for quality evaluation, etc. (Section 4), which are all the points we discuss below in our responses to reviewers). We have also added additional background information on LLJs (notably the different types of LLMs judges and the coverage of our paper) (Introduction and Appendix), along with a new figure summarizing our findings (Section 4).  We have also included our responses to the reviewers questions into our recommendation section (Section 5).

**3-1 Review Response1:**

VTif

**3-2 Reaction To Review1:**

We thank the reviewer for their emergency review and for highlighting the quality and timeliness of our work. We are happy to provide additional context with respect to the weaknesses highlighted. As explained in l.44, we “offer a high-level perspective on the field” as our goal is to provide an overview of the different types of LLJs that are being used, and issues that could emerge. We do not argue that all LLJs applications suffer from all the biases at the same time, rather that the evaluation of LLJs should be context dependent (l.384-387), and we highlight the different issues that could emerge, as for example, position bias/saliency bias/etc. as mentioned in Section 4.2.Robustness. While these biases differ with respect to the type of evaluation, this is not the focus of our work, we rather highlight the lack of robustness of LLJs given all the literature on the topic. Regarding SummEval (l.144-155), what we are trying to show is the fact that LLJ work does not make this distinction between evaluation type of LLJs and that it can create additional problems down the line since each type of evaluation comes with its own biases, as the reviewer also notes (see the Process column in Table 3 that shows how each LLJ paper use its own evaluation type: absolute or pairwise).  The final version of our paper includes additional background on LLJs including the different types of evaluation possible (pointwise, listwise, pairwise) and will help make our contribution more explicit, we will also explicitly mention that the biases differ depending of the type of LLJs earlier in the paper.  Regarding our claim with respect to annotator disagreement, it is very specific to “highly subjective and contested tasks” as explained in the paper l.217-222. In the camera ready, we will mention that annotator disagreement is also used as a proxy for annotation quality, but that it also provides valuable information in subjective tasks (When the Majority is Wrong, Fleisig et al. 2023).

**3-3 Review Response2:**

EPEp

**3-4 Reaction To Review2:**

We thank the reviewer for their thoughtful and positive feedback. We are pleased that the efforts and care invested in this paper are recognized. To answer their questions, other fields (psychology, education, etc.) have spent a lot of time working on what constitutes a good evaluation, and we are happy to see that this literature is being more and more recognized in our community with an increase in papers leveraging measurement theory for NLG evaluation among other things. We believe that ML researchers and practitioners have a lot to learn from this literature, and our goal is to make it more accessible and widely recognized. As experts have already put in the work into investigating what makes a good evaluation. Regarding our recommendations, we believe that various things are needed, some are shorter-term tools to enable more responsible evaluation on a day-to-day basis (model and data cards, checklists, etc.) but other require institutional changes: e.g., Introducing a mandatory course on evaluation in the undergraduate computer science curriculum, alongside changes to incentive structures in publishing that recognize diverse contributions—such as evaluation studies and dataset papers—rather than primarily rewarding model improvement through benchmarking on potentially contaminated datasets.
The camera ready version of our paper will include more details about these suggestions. Additionally, our ongoing research explores how to develop such alternatives to current practices—approaches that are not only cost-effective and scalable, but also enable more thoughtful and responsible evaluation. We believe that responsible evaluation must be context-specific: for each application and development setting of LLJs, distinct challenges may arise. To address this, the community could benefit from tools designed to support their practical use.

**3-5 Review Response3:**

pXGB

**3-6 Reaction To Review3:**

We thank the reviewer for their very nice feedback on our work and for highlighting the rigour and coverage of our work. Regarding their concerns with respect to the environmental impact discussion being tangential to the main arguments, this section comes from this idea of "consequential validity" defined as "the consequences of using the measurements obtained from a measurement model, including any societal impacts."(Jacobs & Wallach, 2021) The societal impacts of measurement models (including environmental impacts) are rarely discussed in our field, and considering that we are transitioning from very efficient measurement tools (BLEU-score and other traditional metrics) to LLMs (that have a considerable environmental impact), we believe it is an important conversation to have. The camera-ready version of our paper will include these additional details to ensure that the argument is better integrated and does not appear isolated.
We have also improved the writing of the paper since the submission (made some sentences shorter) as well as added a figure to summarize our findings to improve the presentation. If the reviewer has additional suggestions, we will make sure to integrate them in the camera-ready version of the paper.
To answer the reviewers question, in the short term, we see value in drawing on insights from experts working on measurement theory to adapt their principles for the evaluation of LLJs. In the longer term, it will be important to consider how evaluation can be better integrated into the curriculum. As computer scientists, evaluation is a central part of our work, yet it is often not adequately addressed in our training.
We also believe that for a successful transition to more responsible practices in the field, we need to develop accessible and cost-effective tools that support researchers and practitioners in their use of LLJs, and we are currently working in this direction.

---

### Meta-Review · Area_Chair_Q9oy · 2025-09-02

**Rating:** 8
**Confidence:** 4

**Strengths:**

This paper analyzes the adoptions of Large Language Models as Judges (LLJs) for evaluating NLG systems and argues that such adoptions have not been fully valid or reliable. The paper examines four assumptions: (1) LLMs as a proxy for human judgment, (2) LLMs as capable evaluators, (3) LLMs as scalable evaluators, and (4) LLMs as cost-effective evaluators. Overall, the paper is:

(1) well motivated and timely. LLJs are increasingly used not only for evaluation but also in LLM training. A position paper that critically reflects on this trend is both necessary and important.

(2) grounded with a principled framework based on measurement theory, providing a systematic way to think about evaluating model capability as evaluators.

(3) well articulated with a comprehensive coverage of LLJ applications and a detailed analysis of the limitations of current practices with concrete examples.

**Weaknesses:**

While this paper has articulated the issues of current practices well, the discussion on how to move forward has not been as thorough.

(1) The discussion could benefit from adding more on how do the authors envision NLG evaluations to be conducted in the future, e.g., what sort of roadmap to follow in order to take steps towards more rigorous evaluations?

(2) Some detailed arguments about the position could be fine-tuned, e.g., claims on that annotator disagreement may be seen as "valuable diversity" could be made more carefully, and that the environmental impact discussion seems somewhat tangential to the main arguments.

The author survey contains some discussions to these points. The next revised version should include the discussions.

**Questions:**

As mentioned above, the authors may add further discussions addressing questions on how to move forward:

(1) How do you envision the field transitioning from current practices? What specific standards or methodologies would you recommend for responsible LLJ validation?

(2) Can we learn from the evaluation practices of interdisciplinary/applied computational work?

The author survey contains some discussions to these points. The next revised version should include the discussions.

**Ethics:**

None.

**Thoroughness:**

4

---

### Decision · Program_Chairs · 2025-09-26

Accept